# Proteomic Profiling of Cardiomyocytes Revealed Potential Radioprotective Effects of Different Resveratrol Pretreatment Regimens

**DOI:** 10.3390/ijms262010223

**Published:** 2025-10-21

**Authors:** Michalina Gramatyka, Marta Gawin, Agata Kurczyk, Adam Gądek, Monika Pietrowska, Piotr Widłak

**Affiliations:** 1Maria Skłodowska-Curie National Research Institute of Oncology, Gliwice Branch, 44-102 Gliwice, Poland; marta.gawin@gliwice.nio.gov.pl (M.G.); agata.kurczyk@gliwice.nio.gov.pl (A.K.); adam.gadek@gliwice.nio.gov.pl (A.G.); monika.pietrowska@gliwice.nio.gov.pl (M.P.); 2Clinical Research Support Centre, Medical University of Gdańsk, 80-210 Gdańsk, Poland; piotr.widlak@nio.gov.pl; 3Maria Skłodowska-Curie National Research Institute of Oncology, 02-781 Warszawa, Poland

**Keywords:** radiation, resveratrol, radioprotection, proteomics, cardiotoxicity, polyphenols

## Abstract

Resveratrol may protect against radiation by modulating cellular metabolism and enhancing the cellular response to stress. Here, we explored its effects on human cardiomyocytes exposed to ionizing radiation. Resveratrol (5 µM) was administered for 1, 7, and 30 days before a single 2 Gy dose of irradiation, and then radiation toxicity and changes in the proteome were evaluated. Extended resveratrol treatment (7 or 30 days) induced more profound proteomic changes than one-day treatment and partially counteracted toxic effects of radiation, leading to increased cell survival, reduced cell death, and fewer cells arrested in the G1 phase. Though resveratrol administration itself had a greater impact on the proteome than radiation alone, we identified three subsets of proteins differently affected by radiation depending on the resveratrol context. The first subset (84 differentially expressed proteins; DEPs) represented proteins influenced by radiation in all resveratrol pretreatment regimens. The second subset (228 DEPs), linked to DNA repair, cell cycle checkpoints, and apoptosis, was affected by radiation only in the absence of resveratrol preconditioning, indicating the compound’s protective effect. The third subset (252 DEPs) involved in metabolism regulation appeared only after extended resveratrol preconditioning. In conclusion, the results demonstrate that hypothetical time-dependent cardioprotective effects of resveratrol are linked to significant proteomic reprogramming of cardiomyocytes caused by long-term pretreatment.

## 1. Introduction

While radiotherapy is a life-saving treatment applied to over half of all cancer patients [1,2,3], the ionizing radiation used to destroy tumors poses a critical long-term health risk to survivors. Furthermore, ionizing radiation is widely used in other diagnostic and medical procedures (e.g., computed tomography, radiography, or treatment of non-malignant diseases) [4,5]. A major consequence of this exposure, especially as patients’ lifespan extends, is radiation-induced cardiotoxicity—the development of cardiovascular diseases years or even decades after treatment [3,6]. The occurrence of cardiovascular diseases, even after exposure to relatively low doses of radiation not exceeding 2 Gy, is a significant concern, partially attributed to radiation’s ability to induce oxidative stress. Such stress leads to energy imbalance (due to mitochondrial damage), altered protein and RNA expression, as well as genetic instability and the release of inflammatory mediators, all of which collectively contribute to the complex pathogenesis of cardiotoxicity [2,7,8,9].

In light of the growing problem of radiation-induced cardiotoxicity, the search for effective radioprotective strategies becomes crucial. Natural compounds, including polyphenols, are of particular interest due to their documented broad spectrum of beneficial biological activities and relatively low toxicity [2,10,11]. Numerous studies indicate that plant polyphenols have the potential for radioprotective action through mechanisms related to enhancing DNA damage repair, scavenging free radicals, increasing the expression of antioxidant enzymes, anti-inflammatory effects, and modulation of cellular pathways involved in metabolism, cell death, and the cell cycle [2,11,12]. One of the most frequently studied polyphenols is resveratrol, a phytoalexin synthesized by plants in response to stress. It demonstrates radioprotective effects, attributed to its ability to scavenge free radicals, inhibit lipid peroxidation, influence the cell cycle, and modulate the transcription of antioxidant enzymes [2,13,14,15,16,17,18,19]. Furthermore, resveratrol is characterized by low toxicity at doses achievable through oral administration (e.g., as a dietary supplement) [14,20] and possesses multiple molecular targets, making it an attractive candidate for a protective compound. Numerous epidemiological and clinical studies indicate that plant polyphenols, including resveratrol, may also beneficially influence the cardiovascular system, improving heart function, microcirculation, and blood parameters, as well as reducing inflammation [12,21,22,23,24,25].

We have previously reported that the effects of a low, dietary achievable dose of resveratrol on the functions of cardiomyocytes depend on the length of exposure to this compound [26]. It should be emphasized that studies on the protective effect of resveratrol, especially using typical in vitro models, often neglect the crucial factors like incubation time and a relevant, physiologically achievable dose. Therefore, this study aimed to investigate the effects of the combined action of ionizing radiation and different resveratrol pretreatment regimens on the molecular and functional response of cardiomyocytes. Based on previous results and available literature, a radiation dose of 2 Gy was chosen as a clinically relevant dose, while 5 µM of resveratrol was chosen as a dose achievable in vivo after oral administration [27,28]. In this study, we report for the first time specific changes in the proteome composition of human cardiomyocytes exposed to both factors, which could contribute to the increased resistance to radiation after resveratrol treatment.

## 2. Results

Our previously published study indicates that exposure of cardiomyocytes to a physiologically achievable concentration of resveratrol (5 µM) alters the activity of signaling pathways related to managing energy metabolism and pro-survival pathways in different time-dependent manners, and that long-term exposure (1 month) resulted in the highest extent of resveratrol-induced effects [26]. Following this observation, we addressed resveratrol-induced changes in proteome profiles of cardiomyocytes conditioned with this compound for 1 day (D.Res), 1 week (W.Res), and 1 month (M.Res) to identify the molecular background of differences observed among these regimens at the functional level. In general, treatment with resveratrol induced massive changes in the proteome composition of cardiomyocytes—there were 1817 differentially expressed proteins (DEPs, large effect size) at either experimental condition vs. untreated control (out of 2828 proteins included in quantitative analyses) (Appendix A). Unsupervised principal component analysis (PCA) revealed distinct clustering of proteomic profiles, with cells treated for 1 week or 1 month clearly separated from untreated controls (Ctr) and cells treated for 1 day (in PC1) (Figure 1A). The number of DEPs was markedly lower after short-term exposure (610 DEPs in D.Res) than after long-term exposures (1597 DEPs in W.Res and 1517 DEPs in M.Res). Importantly, there were only 438 DEPs common for all three conditions, while 1385 DEPs were common for W.Res and M.Res (Figure 1B). Moreover, a positive correlation (r = 0.205) was found between changes induced after 1 week and 1 month of exposure (Figure 1C), which indicated a similar pattern of resveratrol-induced changes after either regimen of extended exposure.

In the next step, the biological pathways associated with subsets of DEPs characteristic of a specific regimen of resveratrol treatment were identified (Appendix A). Among the most abundant protein subsets associated with overrepresented processes were those linked to the regulation of metabolism [HSA-1430728], signal transduction [HSA-162582], immune system [HSA-168256], and cellular response to stress [HSA-2262752]. These processes appeared to be significantly affected after either mode of the resveratrol treatment, despite different subsets of associated DEPs being detected after short-term (1 day) and long-term (1 week/month) treatment (Appendix A). We concluded that though different sets of proteins were affected by different regimens of resveratrol treatment, the major types of processes associated with these proteins were similar. To address processes generally affected by any mode of resveratrol treatment, we focused on DEPs common for D.Res, W.Res, and M.Res, which additionally showed a significant time-dependent trend of resveratrol-induced changes (402 DEPs; Appendix A). Interestingly, in this specific subset of proteins commonly affected by resveratrol, factors associated with regulating cellular signaling (Rho-GTPases pathway and AKT signaling, in particular) and gene expression (including p53-dependent transcription regulation) were identified (Figure 1D). In conclusion, pathways linked to cellular response to stress were commonly activated by all treatment regimens; however, different subsets of resveratrol-affected proteins were linked to this type of cellular response after shorter and extended treatment. This provided a rationale to analyze a hypothetical radioprotective effect of resveratrol in the context of different times of cell preconditioning, which was addressed in the second part of the study.

Cardiomyocytes were irradiated with a single 2 Gy dose after three different regimens of resveratrol preconditioning, one day (IR-D.Res), one week (IR-W.Res), or one month (IR-M.Res), and then different functional tests were applied to address potential protective effects of resveratrol (experimental scheme in Figure 2A; all tests were performed using 6 biological replicas). We noted that irradiation with 2 Gy and short-term treatment with resveratrol increased the metabolic activity of mitochondria, while in cells preconditioned with resveratrol for one month, irradiation did not affect metabolic activity when compared to untreated controls (Figure 2B). Irradiation with 2 Gy significantly (*p* = 0.014) reduced the fraction of clonogenic cells, which was not prevented by a one-day resveratrol pretreatment. On the other hand, extended resveratrol conditioning increased the clonogenic potential of cardiomyocytes, and the number of clones surviving irradiation was significantly higher (*p* = 0.001) after such resveratrol pretreatment compared to radiation alone (Figure 2C). Irradiation with 2 Gy slightly increased the number of cells undergoing apoptosis, which was not reduced by a one-day resveratrol pretreatment; however, the differences did not reach statistical significance (it should be noted that cardiomyocytes are generally resistant to radiation-induced apoptosis and we did not observe a statistically significant increase in the number of apoptotic cells even after a higher dose of 4 Gy). Nevertheless, extended resveratrol conditioning decreased the extent of radiation-induced cell death (*p* = 0.076 for the difference between IR and IR-W.Res). Moreover, irradiation with 2 Gy reduced the number of cells in the S phase and increased the number of cells in the G1 phase (Figure 2D), which resulted in the increased G1:G2/M ratio after irradiation alone (Figure 2E). These changes were more pronounced one hour post-irradiation compared to corresponding effects noted 24 h post-irradiation, suggesting that the studied cells possess a robust capacity to overcome the radiation-induced stress. Resveratrol conditioning increased the number of cells in the S phase and decreased the number of cells in the G1 phase, irrespective of irradiation, and the G1:G2/M ratio in cells irradiated after one month of resveratrol pretreatment was significantly lower (*p* = 0.052) as compared to radiation alone. We concluded that long-term (one week or one month) conditioning of cardiomyocytes with resveratrol partly counteracted the radiation-induced effects: increased survival of clonogenic cells, reduced number of cell deaths, and reduced number of cells arrested in the G1 phase (though not all radioprotective effects were statistically significant, especially regarding the changes in the cell cycle phase distribution). Hence, aiming to elucidate the molecular bases of these potential radioprotective effects of resveratrol, proteome patterns induced upon irradiation and different resveratrol preconditioning modes were analyzed.

Proteomic profiles were analyzed in all experimental variants depicted in Figure 2A. The unsupervised PCA of the proteomics dataset revealed clustering of samples from cells treated with resveratrol for one week or one month, irrespective of irradiation. Similarly, cells not treated with resveratrol or those treated for only one day formed another cluster (in PC1), regardless of irradiation (Figure 3A). A large number of DEPs were identified that differentiated irradiated, non-preconditioned cells (IR) from irradiated cells preconditioned with resveratrol: 694, 1258, and 1194 for IR vs. IR-D.Res, IR vs. IR-W.Res, and IR vs. IR-M.Res, respectively (Figure 3B). Moreover, subsets of DEPs that differentiated untreated cells from resveratrol-treated cells and those that differentiated irradiated cells from cells irradiated after resveratrol preconditioning substantially overlapped. Assuming similar effects induced by long-term conditioning with resveratrol, we identified a large set of 830 DEPs that commonly differentiated resveratrol-untreated cells and cells treated with resveratrol for either one week or one month, both without and after irradiation (Figure 3C). These observations collectively indicated that long-term preconditioning with resveratrol (one week or one month) was the major source of variability in proteomic patterns of cardiomyocytes subjected to the combined treatment.

Preconditioning with resveratrol was the major source of variability in the proteomic patterns of cardiomyocytes subjected to the combined treatment with radiation and resveratrol. Because of this, to identify specific radiation-related effects, we focused on differences between irradiated resveratrol-preconditioned cells and cells with corresponding resveratrol preconditioning alone. Using this approach, we defined distinct subsets of DEPs that were affected by irradiation. Ionizing radiation caused significant changes in the expression of 855 proteins in resveratrol-untreated cells, 810 proteins after one-day preconditioning, 600 proteins after one-week preconditioning, and 527 proteins after one-month preconditioning (Appendix A), indicating that extending preconditioning with resveratrol reduced the number of IR-affected proteins in the time-dependent mode. Moreover, subsets of IR-affected proteins overlapped only partly when different resveratrol preconditioning regimens were considered. There were 228 DEPs affected by IR specifically in the absence of any resveratrol preconditioning, 252 DEPs affected by IR specifically in cells subjected to long-term resveratrol preconditioning, and only 84 DEPs affected by IR in all conditions, regardless of resveratrol preconditioning (Figure 4A).

Proteins significantly affected (upregulated or downregulated) by IR, specifically in cells untreated with resveratrol (or IR-affected in all experimental regimens), were not affected by IR in cells preconditioned with long-term resveratrol treatment (Figure 4B). Furthermore, when we compared levels of IR-affected proteins in cells untreated with resveratrol (IR vs. Ctr) and cells long-term preconditioned with resveratrol (IR-M.Res vs. M.Res), low correlation between both parameters was noted for subsets of IR-affected proteins, either specific for resveratrol-untreated cells or specific for resveratrol-preconditioned cells; in contrast, a strong correlation (r = 0.773) was noted in the subset of proteins affected by IR at any preconditioning regimen (Figure 4C). These observations indicated collectively that preconditioning with resveratrol for either one week or one month largely affected the proteomic pattern in irradiated cardiomyocytes (one-day preconditioning was much less effective). One should conclude that long-term resveratrol preconditioning prevented upregulation/downregulation of several IR-modulated proteins.

In the next step, we performed pathway analysis to identify overrepresented pathways associated with different subsets of IR-affected proteins, depending on resveratrol preconditioning (Appendix A). The first subset included 228 DEPs whose expression was significantly affected by ionizing radiation only in cells not preconditioned with resveratrol (i.e., for these proteins, resveratrol preconditioning moderated/prevented effects of radiation). Among these proteins were those primarily associated with pathways critical for the toxicity of radiation: cellular stress response, regulation of the cell cycle, DNA repair, and cell death (Figure 4D). In contrast, the subset of 252 DEPs affected by radiation only in cells preconditioned with resveratrol for one week or one month included proteins primarily associated with the regulation of metabolism, while proteins associated with response to stress represented a minor fraction (Figure 4E). Notably, this subset included proteins modulated by long-term resveratrol treatment specifically, whose resveratrol-induced changes were somehow reversed by radiation, e.g., among 73 DEPs downregulated by radiation specifically in cells preconditioned by resveratrol for one month (i.e., IR-M.Res vs. M.Res), there were 60 proteins up-regulated by one month of resveratrol treatment itself (i.e., M.Res vs. Ctr) (Appendix A). A smaller subset of 84 DEPs that were affected by radiation at any experimental condition (regardless of resveratrol treatment) includes proteins associated with cellular stress response, the cell cycle, and apoptosis (i.e., typical radiation-responsive pathways) (Figure 4F). A large fraction of IR-affected proteins has a putative nuclear localization (GO:0005634): 58% of 228 DEPs specific for no-resveratrol conditions, 69% of 84 DEPs common for all conditions, and 53% of 252 DEPs specific for long-term resveratrol preconditioning. Such nuclear localization of IR-affected proteins could be expected, assuming the cellular localization of critical processes linked to the toxicity of ionizing radiation. In comparison, the majority of proteins affected by resveratrol treatment have putative cytoplasmic localization (GO:0005737), from 81% to 86%, depending on the treatment regimen (Appendix A). Our results showed collectively that long-term preconditioning with resveratrol has a large impact on the response of cardiomyocytes to ionizing radiation, which was reflected by the prevention (or modulation) of IR-related changes in the expression of proteins linked to processes essential for the toxic effects of radiation: the cell cycle, DNA repair, and cell death.

## 3. Discussion

The increasing application of ionizing radiation in cancer therapies and other medical procedures necessitates the search for radioprotectants: substances that can counteract the harmful effects of radiation observed, among others, in the form of cardiotoxicity. Currently, only two radioprotective compounds, amifostine and palifermin, have been approved for use, and frequent adverse side effects hinder their application [2]. For this reason, numerous studies focus on searching for novel, preferably natural products, available in the form of diet ingredients that could be used to reduce radiation toxicity. This study aimed to evaluate whether a physiologically achievable concentration of resveratrol (5 µM) can influence the toxicity of ionizing radiation in cardiomyocytes, and how this effect depends on the duration of cells’ preincubation. Cardiomyocytes were chosen due to the known cardiotoxicity of radiation, and the use of relatively low dose (2 Gy) is significant, as even minimal exposure can increase the risk of cardiovascular diseases, a fact confirmed in the literature [29].

Our previous results documented that the influence of resveratrol on the phenotype of cardiomyocytes is highly dependent on the duration of exposure [26]. We observed that the longer the preincubation time, the more pronounced the effect induced by resveratrol. Here, we extended this observation by the analysis of resveratrol-induced molecular reprogramming of cardiomyocytes. The proteomic analysis revealed that resveratrol alters pathways related to energy metabolism, cell survival, signal transduction, stress response, and immune response. These observations are consistent with existing literature, which describes resveratrol’s ability to modulate energy metabolism and exert anti-inflammatory effects. The metabolic modulation is often linked to the substance’s capacity to stimulate AMPK and mTOR pathways [2,16,30,31], while its anti-inflammatory properties may be the result of the suppression of NF-kB activation [3,31]. Unfortunately, the untargeted proteomic analysis used in the current study did not allow for the identification of all proteins relevant to the aforementioned pathways. For example, transcription factors of the NF-κB family were not detected; nevertheless, the upregulation of AKT1 was confirmed, along with several proteins associated with the mTOR complex. However, resveratrol-mediated changes in the activity of these crucial pathways were documented in our previous study [26] using Western blotting. The modulation of NF-kB may be connected to the observed impact of resveratrol on the stress response and gene expression. The mechanism of this action is often attributed to Sirt1 in the literature [31,32,33,34]. However, our study did not show any direct changes in Sirt1 in cardiomyocytes [26] and present study. Hence, there is a possibility that resveratrol does not alter Sirt1 expression but rather its activity, which would require further studies. It is also plausible that at the concentration used in the current study, resveratrol does not activate Sirt1 [31]. In this case, the observed changes would have another source, possibly through the regulation of cAMP content [31]. This effect could be linked to the putative “phytoestrogen” activity of resveratrol and its ability to activate estrogen receptors [35,36,37], which, in addition to controlling the reproductive system, affects inflammatory processes by influencing the NF-κB pathway and secretion of cytokines [38]. Moreover, it has been postulated that resveratrol-induced activation of the PI3K/Akt pathway could also be mediated by estrogen receptors [38]. Notably, our proteomic data revealed that pathways related to estrogen-dependent gene expression and estrogen signaling were also affected after exposure to resveratrol.

It is generally postulated that resveratrol-induced metabolic reprogramming plays a critical role in a hypothetical radioprotective activity of this compound, especially in the context of counteracting oxidative stress and impaired energy metabolism [39]. Therefore, exposure duration-dependent effects of resveratrol on molecular reprogramming of target cells observed in our study provided a new dimension to the mechanism of potential radioprotective activity of this compound. To investigate this mechanism, we irradiated cardiomyocytes pre-conditioned with resveratrol for different durations. We found that long-term (one week or one month) conditioning of cardiomyocytes with resveratrol partly counteracted the radiation-induced effects. This was evidenced by increased survival of clonogenic cells, a reduced number of cell deaths, and a decrease in the number of cells arrested in the G1 phase. Though only part of the observed differences were statistically significant (while other data should be considered a trend), all functional data were coherent, indicating that extended preconditioning with resveratrol had radioprotective effects. Therefore, the initial finding that longer treatment with resveratrol had a more profound effect on the metabolism of cardiomyocytes was in line with the observation of more effective radioprotection. Several studies documented the radioprotective action of resveratrol. Treatment with resveratrol reduced the number of chromosomal aberrations [20] and overall radiation-induced DNA damage [40,41,42,43]. Moreover, the increase in cell viability, proliferation, and promotion of DNA repair [42,44], as well as enhanced activity of antioxidant enzymes [44] and changes in energy metabolism [45] in irradiated cells and tissues after resveratrol administration were also reported. However, putative duration-dependent effects of resveratrol in the context of radioprotection have not been addressed systematically. Only a few reports concerned this aspect of resveratrol treatment, including the report of Takahashi and coworkers, who noted positive effects of resveratrol on the vascular system only after longer (more than 5 days) treatment [46].

In this study, a comprehensive analysis of the proteome of irradiated cardiomyocytes pre-conditioned with resveratrol using different times (corresponding to both acute and chronic/persistent treatment) was performed for the first time, which allowed us to address the molecular reprogramming of cells exposed to the combined action of both factors. Our analysis revealed that exposure to resveratrol, especially the extended pre-conditioning, had a markedly greater impact on the cardiomyocyte proteome profile than the irradiation itself. This indicates a completely different background for radiation-induced effects in pre-conditioned and “naïve” cells, emphasizing the significance of persistent exposure typical of hypothetical dietary supplements. The pairwise comparisons between irradiated resveratrol-preconditioned cells and cells with corresponding resveratrol preconditioning alone allowed us to assess the radiation-induced changes per se. Importantly, three subsets of IR-affected proteins were identified, which corresponded to three types of responses to the combined action of resveratrol and radiation. The first subset included proteins that were similarly affected by radiation both in the absence of resveratrol-pretreatment and after any regimen of resveratrol preconditioning. This subset of proteins represented the core response to radiation and included proteins associated with basic processes involved in the cellular response to radiation-induced damage, including apoptosis and the cell cycle checkpoints. This aligns with the observed functional impairments following irradiation, including reduced clonogenic potential, as well as disturbances in cell cycle progression and apoptotic intensity. The second subset included proteins that were markedly affected by radiation only in cells that were not pre-conditioned with resveratrol at all, which indicated that resveratrol-pretreatment prevented their IR-related changes. This subset included proteins associated with mechanisms critical for radiation-induced toxicity: DNA repair, regulation of the cell cycle, and programmed cell death. Notably, the same processes were affected phenotypically due to the extended resveratrol preconditioning when testing the hypothetical radioprotective effects of this compound. Therefore, this specific subset of proteins could be considered as a molecular footprint of the radioprotective activity of resveratrol. One should be aware that proteins found in this subset are not necessarily associated with causative mechanisms of resveratrol-mediated radioprotection but could rather reflect radioprotective effects of such treatment (e.g., reduced apoptosis and normalized cell cycle progression). The third subset included proteins for which IR-induced changes in abundance were detected only after the extended pre-conditioning with resveratrol (i.e., were not affected by radiation in naïve cells or cell preconditioned for one day). The most abundant groups of proteins in this subset were associated with the metabolism of RNA and proteins, yet only a few were related to the response to stress. Hence, this subset reflected an interesting metabolic aspect of the interplay between both factors, yet without the direct influence of typical processes associated with radiation-induced damage. Notably, it seemed that to some extent, radiation reversed the effects of the extended resveratrol treatment (e.g., downregulating proteins upregulated by resveratrol itself). Taken together, observed changes in proteomic profiles suggest that while extended resveratrol exposure induces adaptive changes in cellular metabolic processes that do not significantly affect basal viability, the resulting molecular reprogramming is essential to the protective effect of resveratrol observed following irradiation.

Proteomic changes induced by ionizing radiation in cells with metabolism differently reprogrammed by acute and persistent preconditioning with resveratrol provide a picture of a molecular background for a hypothetical radioprotective activity of this compound. Our data cannot be directly compared with other proteomics datasets due to the lack of relevant reports. Nevertheless, available literature documents the overlap of molecular processes affected by resveratrol and ionizing radiation. In general, irradiation of cells and tissues results in increased oxidative stress and inflammation, dysfunctions related to energy metabolism and signal transduction, as well as activation of stress response, cell-cycle control, and DNA repair pathways. These effects are observed at the transcriptomic [47,48,49], proteomic [50], and metabolomics [51,52] levels. On the other hand, the opposite effects induced by resveratrol were reported, including improved energy metabolism [44,53,54] as well as reduced inflammation [30,55] and oxidative stress [30,54].

Despite utilizing clinically relevant doses of radiation and physiologically achievable concentrations of resveratrol, it is crucial to recognize the inherent limitations of this in vitro model. While our findings offer valuable molecular insights into the potential radioprotective mechanisms of resveratrol, they cannot be directly extrapolated to clinical settings. Nevertheless, the current results provide a strong foundation for designing a follow-up animal study. Such an in vivo investigation could both confirm the clinical utility of long-term resveratrol supplementation for radioprotection and allow for a deeper exploration of systemic mechanisms, particularly those related to the immune system and the inflammatory state, which cannot be analyzed in simple cellular models.

## 4. Materials and Methods

Cell line and materials. Human cardiomyocytes were obtained from Celprogen (Torrance, CA, USA; catalog number 1311001-09) and cultured in flasks coated with fibronectin. Cells were cultured in DMEM/F12 medium supplemented with 10% FBS, 2 mM L-glutamine, and antibiotics (all from Corning, Tewksbury, MA, USA), in 37 °C humidified atmosphere containing 5% CO_2_. Trans-resveratrol was dissolved in DMSO (both from Sigma Aldrich, Burlington, MA, USA), and its stock solution (100 mM) was stored at −20 °C in the dark. Cells were incubated with 5 µM resveratrol for 1 day (24 h), 1 week (7 days), or 1 month (30 days) before irradiation, and then irradiated with a 2 Gy radiation dose. Irradiation of cells was performed using TrueBeam linear accelerator (Varian, Palo Alto, CA, USA) with 6MV photons. Each irradiation plan was calculated for the corresponding dose with one beam of 300 MU/min dose rate and an SSD of 100 cm.

Colony-forming assay. Cells were seeded into 6-well plates at a density of 1000 cells per well, irradiated, and then cultured in fresh medium for 10 days. The colonies were stained with a solution containing 0.5% crystal violet and 50% methanol, rinsed with water to remove excess dye, and then the number of colonies was counted.

Cell viability assay. The XTT assay was used to determine cell viability and metabolic activity. Cells were seeded into 96-well plates. 24 h after irradiation, XTT reaction solution (Biological Industries, Kibbutz Beit Haemek, Israel) was added, plates were incubated in an incubator (37 °C, humidified atmosphere, 5% CO_2_) for 3 h, and absorbance was measured at wavelengths 470 nm and 630 nm.

Cell cycle analysis. To characterize a distribution of the cell cycle phases, cells were harvested by trypsinization and fixed with 70% ethanol 1 h or 24 h after the irradiation. In collected cells, DNA was stained with PI/RNase Staining Buffer (BD Biosciences, Franklin Lakes, NJ, USA), and then analyzed by flow cytometry using a FACSCanto cytometer (Becton Dickinson, Franklin Lakes, NJ, USA).

Apoptosis analysis. To assess the intensity of cell death, the FITC Annexin V Apoptosis Detection Kit (BD Pharmingen, Franklin Lakes, NJ, USA) was used. In brief, 24 h after irradiation, cells were harvested by trypsinization, washed twice in PBS, and suspended in Binding Buffer. FITC Annexin V and propidium iodide were added to the cell suspensions and incubated in the dark for 15 min. Samples were analyzed by flow cytometry using a FACSCanto cytometer (Becton Dickinson, Franklin Lakes, NJ, USA).

Proteomic analysis. Whole-cell lysates in RIPA buffer (1% NP-40, 0.5% SDC, 0.1% SDS with protease and phosphatase inhibitors) were prepared for mass spectrometry-based proteomic analysis using the FASP (Filter-Aided Sample Preparation) protocol, as described by Wisniewski et al. [56], 48 h after irradiation. Briefly, 50 µg of protein in each sample was loaded into a spin ultrafiltration unit (30 kDa cut-off; Merck, Darmstadt, Germany), and proteins retained on the filtering membrane were purified from TLB lysis buffer by repeated washes with 8M urea in 0.1M Tris-HCl, pH 8.5, followed by alkylation with iodoacetamide (0.05M). Then, samples were digested in trypsin solution (enzyme to protein ratio 1:100, *w*/*w* at 37 °C for 18 h), peptides were fractionated into two fractions using the SAX (Strong Anion Exchange) technique, and desalted on C18 tip columns. Each sample was divided into two peptide fractions, eluted at pH 5 and 2. The concentration of obtained peptides was determined by tryptophan fluorescence. Individual fractions were analyzed with the use of Dionex UltiMate 3000 RSLC nanoLC System connected to Q Exactive Orbitrap mass spectrometer (Thermo Scientific, Waltham, MA, USA). Peptides from each fraction (1 μg) were separated on a reverse-phase Acclaim PepMap RSLC nanoViper C18 column (75 μm × 50 cm, 2 μm granulation; Thermo Scientific, Waltham, MA, USA) using an acetonitrile gradient (from 3 to 35% for 220 min, in 0.1% formic acid) at 35 °C and a flow rate of 300 nL/min (for 240 min). The spectrometer was operated in data-dependent MS/MS mode with survey scans (top 12) acquired at a resolution of 70,000 at *m*/*z* 200 in MS mode in the scanning range of 380–1500 *m*/*z*, and 17,500 at *m*/*z* 200 in MS2 mode. Spectra were recorded in the scanning range of 200–2000 *m*/*z* in the positive ion mode. Higher energy collisional dissociation (HCD) ion fragmentation was performed with normalized collision energies set to 28. Protein identification was performed using the Swiss-Prot human database with a precision tolerance of 10 ppm for peptide masses and 0.02 Da for the fragment ion masses. All raw data obtained for each dataset were imported into Proteome Discoverer v.2.3 (Thermo Scientific) for protein identification and quantification. Fractions derived from each sample were combined before analysis, and then the Sequest engine was used for database searches. Protein was considered positively identified if at least two peptides per protein, and a peptide score reached the significance threshold FDR = 0.01 (assessed by the Percolator algorithm); a protein was further considered as “present” if detected in at least one sample of a given type. Abundances of identified proteins were estimated using Proteome Discoverer using the Precursor Ions Area detector (Thermo Scientific, Waltham, MA, USA) node with further normalization to the total ion current chromatogram. Normalized data was exported to an MS Excel file for further statistical analysis.

Statistical and bioinformatics analysis. All analyses were performed using at least three independent experiments (6 biological replicas were included in functional tests). Significance of differences between compared groups in functional tests was assessed by the Kruskal–Wallis test, followed by Dunn’s multiple comparison test. *p*-values below 0.05 were considered statistically significant. For the proteomics dataset, values below the detection limit were considered missing and imputed using random draws from a lognormal distribution. The parameters of this distribution were estimated by maximum likelihood, accounting for left truncation between zero and the lowest observed non-missing value for each protein (proteins with ≥50% missing values in any group were excluded from quantitative analyses). For the quantitative dataset (2828 proteins with <50% missing data per group), group differences in protein abundance were assessed using the Kruskal–Wallis test. Post hoc pairwise comparisons were performed with the Conover test [57]. Effect sizes were quantified by eta-squared for the Kruskal–Wallis test, while Conover test statistics were standardized by the square root of the sample size to obtain Pallant’s r, interpreted according to Cohen’s thresholds [58,59]: negligible (|r| < 0.1), small (|r| ≥ 0.1), medium (|r| ≥ 0.3), and large (|r| ≥ 0.5). To test for ordered differences across the four experimental conditions (Ctr, D.Res, W.Res, M.Res), the Jonckheere–Terpstra test was applied [60]. All hypotheses were tested two-sided at the 5% significance level. Multiple comparisons were adjusted using the Benjamini–Hochberg procedure, with adjusted *p* < 0.05 considered significant. Analyses were conducted in R (version 4.4.1). The STRING database [61] was used to estimate the overrepresentation of associated REACTOME pathways and to illustrate potential relationships between selected proteins (https://string-db.org/, accessed on 10 September 2025).

## 5. Conclusions

In this work, we provide evidence that resveratrol-mediated metabolic reprogramming of cardiomyocytes depends on the duration of resveratrol conditioning, which adds dimension to the mechanisms of potential radioprotective activity of this compound. Diminished toxic effects of ionizing radiation were noted in cells irradiated after extended conditioning with physiologically relevant doses of resveratrol, which hypothetically mimicked its usage as a dietary supplement. Furthermore, proteomics profiling identified a molecular signature of the radioprotective activity of resveratrol that consisted of proteins associated with DNA repair, the cell cycle checkpoint, and cell death, which radiation-related deregulation was prevented in cells preconditioned with resveratrol. However, due to the limitations of the in vitro model applied, the obtained results could only be used as proof-of-concept and need to be further validated in vivo.

## Figures and Tables

**Figure 1 ijms-26-10223-f001:**
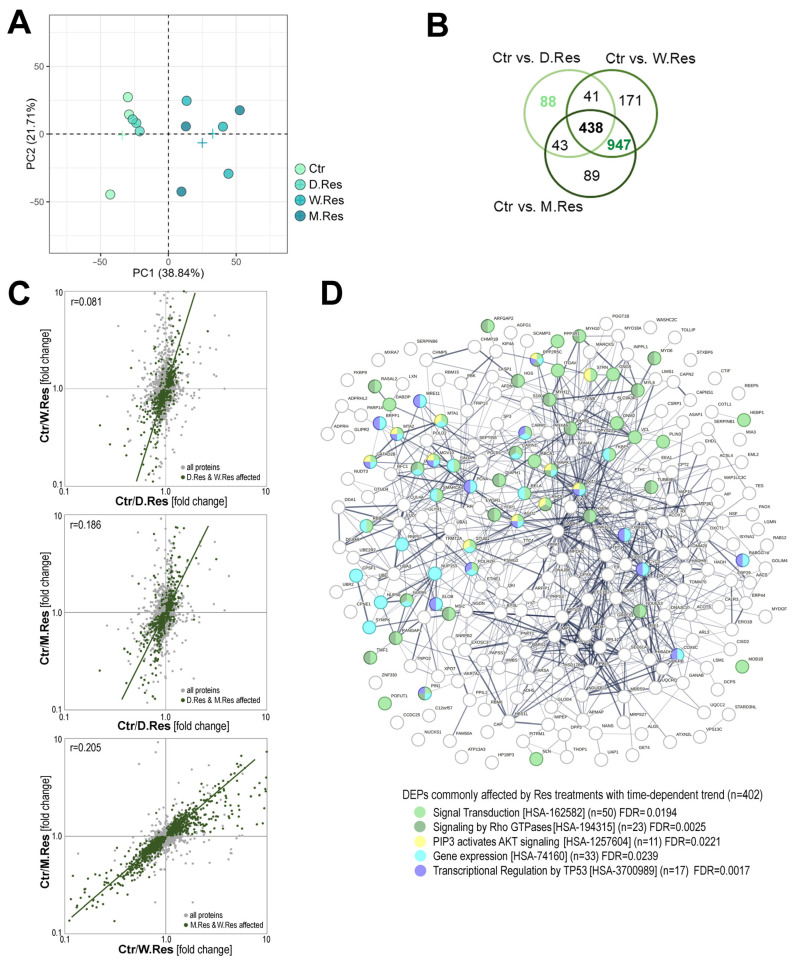
Resveratrol-induced changes in the proteome profiles of cardiomyocytes depend on treatment regimens. Panel (**A**)—Principal component analysis (PCA) showing overall similarities between samples of cells treated with 5 µM resveratrol for 24 h, 7 days, and 30 days (D.Res, W.Res, and M.Res, respectively); a cross represents the average for each group. Panel (**B**)—Venn diagram representing the overlap of Differentially Expressed Proteins (DEPs; with at least a large effect size) between untreated controls and cells pretreated with resveratrol for varying time periods. Panel (**C**)—Pairwise correlations of resveratrol-induced changes in protein abundance (fold change Ctr vs. Res) between different time regimens; shown are correlation coefficients, with DEPs common to both regimens indicated. Panel (**D**)—Functional networks of proteins that were commonly affected by different regimens of resveratrol treatments (overlapping DEPs with significant time-related trend; the Jonckheere–Terpstra trend test *p* < 0.05); proteins associated with the selected REACTOME pathways are color-coded, along with the significance of the process overrepresentation.

**Figure 2 ijms-26-10223-f002:**
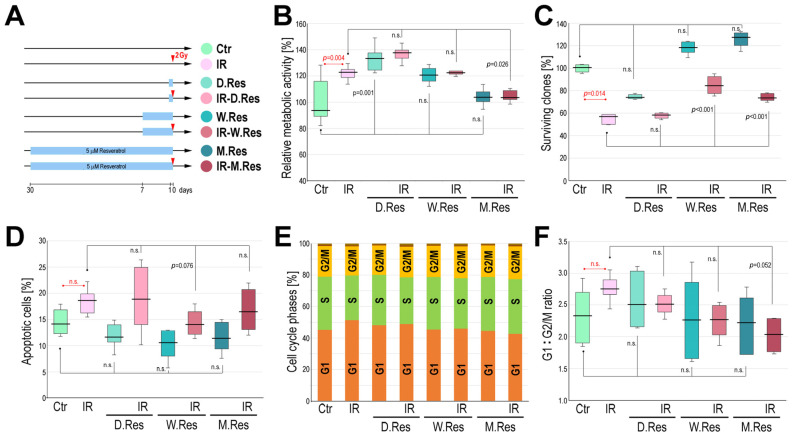
Effects of cardiomyocytes’ pre-treatment with resveratrol on response induced by ionizing radiation. Panel (**A**)—Diagram of the experimental scheme. Panel (**B**)—The influence of irradiation and resveratrol preconditioning on relative metabolic activity of cardiomyocytes (in relation to the average in untreated control). Panel (**C**)—The influence of irradiation and resveratrol preconditioning on the number of surviving clones (in relation to the untreated control). Panel (**D**)—The influence of irradiation and resveratrol preconditioning on the fraction of cells undergoing apoptosis. Panel (**E**)—The influence of irradiation and resveratrol preconditioning on the distribution of the cell cycle phases 1 h after irradiation. Panel (**F**)—The influence of irradiation and resveratrol preconditioning on the ratio between cells in G1 and G2/M phases of the cell cycle 1 h after irradiation. Boxplots on (**B**–**D**,**F**) represent median, upper and lower quartiles, maximum, and minimum. Showed is statistical significance of differences between untreated control and irradiated cells (red symbols), untreated control and resveratrol-treated groups, as well as cells irradiated without resveratrol preconditioning and groups irradiated after different preconditioning (n.s.—not significant; *p* > 0.1).

**Figure 3 ijms-26-10223-f003:**
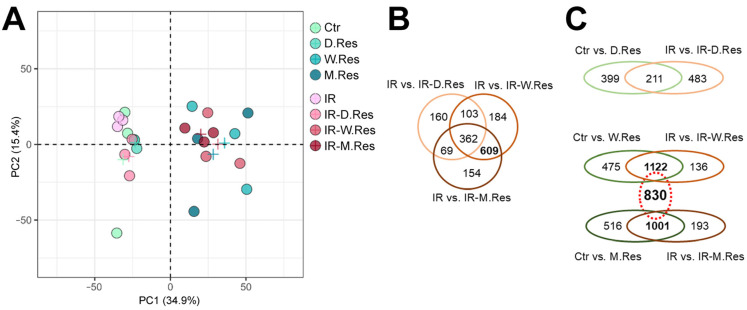
Combined exposure to resveratrol and ionizing radiation affects proteome profiles in cardiomyocytes. Panel (**A**)—Principal component analysis (PCA) showing overall similarities between samples co-treated with ionizing radiation and resveratrol; a cross represents the average for each group. Panel (**B**)—Venn diagrams representing the overlap of subsets of DEPs (with at least a large effect size), differentiating cells irradiated with and without resveratrol preconditioning. Panel (**C**)—Venn diagrams representing the overlap of subsets of DEPs affected by resveratrol treatment with or without radiation (dotted oval represents DEPs common for one-week and one-month resveratrol treatment).

**Figure 4 ijms-26-10223-f004:**
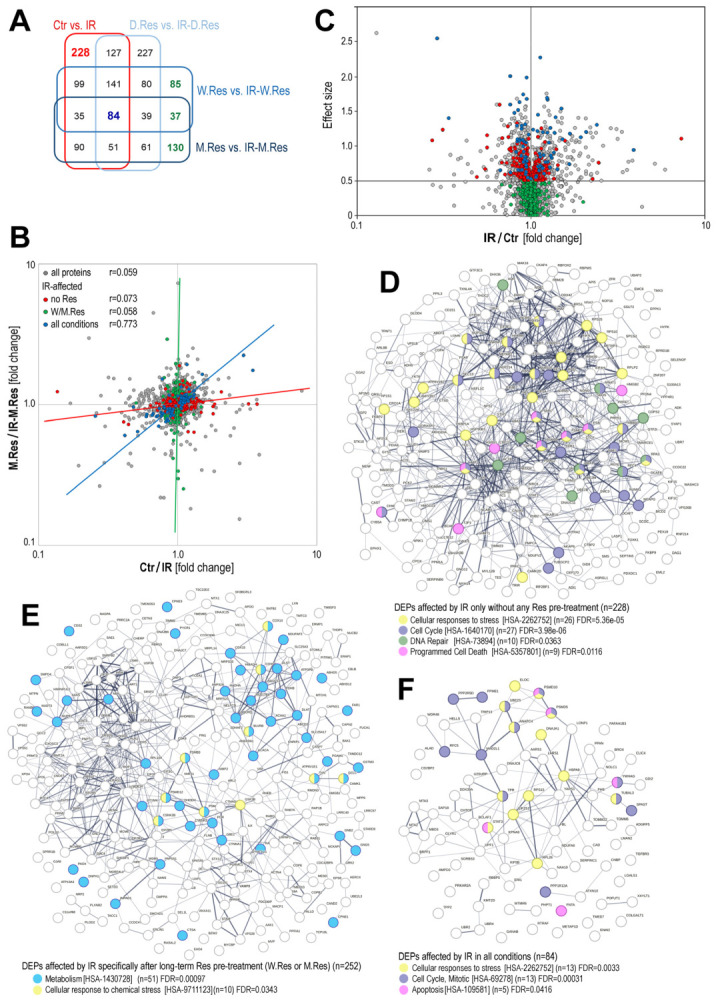
Radiation-induced changes in proteome profiles in resveratrol-preconditioned cardiomyocytes. Panel (**A**)—Venn diagram representing the overlap of subsets of DEPs (at least a large effect size), differentiating irradiated and non-irradiated cells. Panel (**B**)—Volcano plot illustrating the significance of IR-induced changes in cells not preconditioned with resveratrol. Marked are DEPs affected by IR only in not preconditioned cells (red), IR-affected in groups irrespective of preconditioning schemes (blue), and IR-affected only in cells preconditioned with long-term resveratrol treatment (W.Res and/or M.Res; green). Panel (**C**)—Correlation of changes induced by irradiation in not-preconditioned cells and in cells preconditioned with a month-long resveratrol treatment (compared to resveratrol alone); marked are the same subsets of DEPs as in Panel (**B**). The functional networks of IR-affected DEPs are specific to cells not preconditioned with resveratrol (Panel (**D**)), specific to cells irradiated after long-term preconditioning (Panel (**E**)), and common to all groups irrespective of resveratrol treatment (Panel (**F**)). Proteins associated with the selected REACTOME pathways are color-coded, along with the significance of the process overrepresentation.

## Data Availability

The mass spectrometry proteomics data have been deposited to the ProteomeXchange Consortium via the PRIDE [62] partner repository with the dataset identifier PXD069084 and 10.6019/PXD069084 (https://www.ebi.ac.uk/pride/ (accessed on 2 October 2025)). The raw data supporting the conclusions of this article will be made available by the authors on request.

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
