# Peer review of "Proteomic Profiling of Cardiomyocytes Revealed Potential Radioprotective Effects of Different Resveratrol Pretreatment Regimens"

_ijms, 2025, doi:10.3390/ijms262010223_

Round 1
Reviewer 1 Report
Comments and Suggestions for Authors
The manuscript "Proteomic Profiles in Cardiomyocytes Exposed to Ionizing Radiation and Resveratrol: Potential Radioprotective Mechanisms
of the Different Resveratrol Pretreatment Regimens" addresses an important topic in oncology as radiation-induced heart disease represents a growing problem. Understanding how to protect cardiomyocytes is clinically significant, and the potential use of a naturally polyphenol, resveratrol, represents a promising strategy with minimal side effects.
The research design is generally appropriate, with irradiation doses relevant to radiotherapy and resveratrol concentrations that can be realistically achieved through dietary supplementation. The study also models both short-term and chronic resveratrol intake, which reflects real-world patterns of natural supplement use.
The authors applied several complementary methods to evaluate the radioprotective effect of resveratrol on cardiomyocytes and the methodology is overall well described. However, some aspects of the experimental design warrant further clarification. For cell cycle analysis, the 1h post-irradiation timepoint is very early and may miss checkpoint activation, which typically occurs later. A later analysis (e.g., 12–24h post-irradiation) would likely provide more informative results and could be better correlated with apoptosis and viability assays. The study is based on a single in vitro cardiomyocyte model. While this represents a limitation in terms of physiological relevance, it is understandable given the long-term resveratrol incubation and irradiation protocols, which would be challenging to apply in more complex models. Nonetheless, the authors should acknowledge this limitation in the discussion.
The authors report that the irradiation significantly reduces the fraction of clonogenic cells, increases apoptosis, and reduced the number of cells in S-phase. However, the figures only show significant difference for clonogenic survival. While usually the irradiation does affect cell cycle and induce apoptosis in cell culture, the effects are dependent on radiation dose and time following exposure, and in this case only using one relatively small dose and sort time following irradiation (1h for cell cycle analysis and 24h for apoptosis staining) may be the reason for the apparent lack of response. The authors should justify their choice of a 1h timepoint for cell cycle analysis, as checkpoint activation and arrest often occur several hours after irradiation. Additional timepoints (e.g.: 24h) could provide a more complete picture of the cell cycle response.
The XTT test is used here as a measurement of relative metabolic activity, showing slight increase in cellular activity after irradiation. These time of test based on formazan derivates are usually known as viability assays. considering the short time following irradiation, the test indeed shows more likely metabolic activity, and the increase can represent a temporary response necessary for cellular repair/signaling.
Concerning the effect induced by resveratrol alone, Figure 2, C shown that the compound did not affects cellular survival in the absence of irradiation. The figure however shown a notable difference between control and resveratrol treatment (~70% for 1 day treatment, and ~120% for 1 week and 1 month treatment ) with small error bars. The statistical analysis should be carefully checked.
Concerning the metabolic effect of resveratrol alone, there is a notable increase in XTT signal after a short 1 day application, a lower, non-significant increase after 1 week incubation, and the metabolism comes back to normal after chronic 1 month incubation with the polyphenol, in line with the previous observation that metabolic activity represents a time-dependent parameter.
The authors mention that resveratrol conditioning increased the number of cells in S-phase and decreased the number of cells in G1 and G1:G2/M ratio and concludes that long term conditioning partly counteracted the radiation-induced effects. However, the effects are not statistically significant.
Proteomic analysis reveals for the first time here, the pathways involved in cellular response to radiation and/or resveratrol effect in cardiomyocytes, highlighting some key molecular pathways involved in energy metabolism, DNA repair pathways, cell cycle regulation, cellular death mechanisms. While the analysis bring relevant and novel insights, the results would benefit from a more integrated discussion that links cell survival, apoptosis, and proteomics findings, rather than presenting them as separate outcomes.
Overall, this study addresses a timely and clinically relevant question in cardio-oncology and provides novel data on the potential radioprotective role of resveratrol in cardiomyocytes. The experimental design is generally sound, and the proteomic analysis adds value. However, some aspects of the methodology (particularly the choice of timepoints), interpretation of cell cycle/apoptosis results, and the depth of proteomic discussion require further clarification. Addressing these issues would considerably strengthen the manuscript. I therefore recommend major revision before the paper can be considered for publication.
Reviewer 2 Report
Comments and Suggestions for Authors
Dear Authors,
the manuscript entitled "Proteomic Profiles in Cardiomyocytes Exposed to Ionizing Radiation and Resveratrol: Potential Radioprotective Mechanisms of the Different Resveratrol Pretreatment Regimens" investigates the proteomic response of human cardiomyocytes exposed to resveratrol, with a focus on oxidative stress, DNA repair, and metabolic regulation. The study addresses a relevant topic, as resveratrol has been extensively studied for its cardioprotective properties but its molecular mechanisms remain only partially understood. The experimental design is generally appropriate, and the integration of proteomic data provides valuable insights. However, several aspects of data interpretation, methodological clarity, and discussion require revision to improve the robustness and readability of the manuscript. In particular: Abstract (Lines 11-27): the description of the three protein subsets should be simplified to improve readability.
Introduction: Recent references on the clinical cardioprotective effects of resveratrol are missing. The novelty of the study (proteomic analysis in human cardiomyocytes and exposure duration) should be emphasized more clearly.
Lines 265-374: The discussion is overly descriptive and sometimes speculative. The role of Sirt1 is interpreted without direct evidence. A critical analysis of the model limitations (in vitro cells, not adult cardiomyocytes) is missing. A paragraph on translational perspectives and the real bioavailability of resveratrol in vivo should be included.
Supplementary Materials (Tables S1-S3, Fig. S1): The tables are difficult to navigate; reorganizing them into a well-structured Excel file with separate sheets would improve usability. Figure S1 has low resolution and should be improved for better readability.
(Lines 460-469): The conclusions are too optimistic. A statement acknowledging experimental limitations and the need for in vivo validation should be added.
Author Response
Please see the attachment.
Please note that high-resolution images (600 dpi) were uploaded as separate files.

Reviewer 3 Report
Comments and Suggestions for Authors
The study presents a novel approach by analyzing the duration of pretreatment with resveratrol in irradiated cardiomyocytes. The experimental design is suitable for proteomic analysis but lacks strength in functional validation. The overall consistency of the manuscript is adequate, although it would be advisable to reorganize the results to give more weight to the most important findings of the study. Below are my observations.
1. It is recommended that the title be simplified and the phrase "Potential Radioprotective Mechanisms" be abbreviated to avoid redundancy.
2. It is recommended that the abstract be improved by including key numerical values, and its conclusion should also be improved.
3. It is suggested that synonyms or mechanistic terms be included in the keywords, and that the exact words from the title not be repeated to achieve better indexing.
4. Improve the first paragraph of the introduction to make it more accessible to an audience unfamiliar with the subject (this paragraph should capture the reader's attention). The background is well contextualized but too lengthy (it is recommended to summarize it); the research gap and the research hypothesis should be made explicit. The specific objectives should be stated as concrete statements at the end of the introduction (a common practice is to do this as bullet points).
5. The discussion is only comparative, but not sufficiently critical of the limitations of extrapolating in vitro to clinical conditions. The discussions should be improved by emphasizing the biological, physical, and chemical mechanisms involved. Include more current citations to strengthen the discussions.
6. Mechanisms (NF-κB, Sirt1, PI3K/Akt) are mentioned, but without direct evidence in this study; the critical discussion of this part should be improved.
7. More in-depth mechanistic explanations of the connection between DEPs and the observed phenotype are lacking.
8. There is no discussion of cost analysis, scalability, or clinical applicability, as recommended by IJMS in translational studies.
9. In the materials and methods, the cell line, media, and supplements are described, but the degree of purity of reagents and the exact origin (brand, city, country) are missing. The equipment is mentioned (e.g., TrueBeam, Orbitrap), but the complete model and manufacturer are not provided in all cases. Take into account the MDPI guidelines.
10. The experimental design should be explicitly stated (it is not explained with a logical diagram or the assumptions that support it), nor is it indicated whether the experiments were independent in biological and technical replicates.
11. The imputation of proteomic data is described, but it is advisable to justify why this method was chosen over others.
12. The conclusions do not mention limitations (e.g., cell model, single radiation dose, lack of in vivo validation). Clear prospects, such as validation in animal models or clinical studies, are not included.
13. Use the ENDNOTE PLUGIN to cite correctly.
14. It is recommended to check the similarity index and paraphrase it, as according to the Ithenticate report, it is 22%.
In English, there are long and redundant sentences that can be simplified for clarity. It would be advisable to have a native editor review the text for fluency.
Author Response

(The authors gave the same response as above.)

Reviewer 4 Report
Comments and Suggestions for Authors
This review is prepared for the article titled Proteomic Profiles in Cardiomyocytes Exposed to Ionizing Radiation and Resveratrol Potential Radioprotective Mechanisms of the Different Resveratrol Pretreatment Regimens. The study provides a comprehensive analysis using proteomic profiling to clarify how resveratrol (RSV) pre-conditions cardiomyocytes and protects them from ionizing radiation (IR), emphasizing the significance of the pre-incubation time.
The primary area for improvement concerns the statistical ambiguity of key functional results. Although a clear protective trend was observed following long-term treatment, not all radioprotective effects achieved statistical significance. For instance, the reduction in apoptosis after IR-W.Res pre-conditioning 1 week showed a p-value of 0.076, which is borderline significant. Similarly, the reduction in the G1G2/M ratio after IR-M.Res pre-conditioning 1 month had a p-value of 0.052, which is also borderline. The authors should strengthen these functional conclusions potentially by increasing the number of replicates n in the biological experiments or by adjusting how p-values close to 0.05 are interpreted, treating them perhaps as statistical trends rather than fully significant conclusions. Another important issue is the ambiguity surrounding the Sirt1 mechanism. The discussion references literature linking resveratrol’s action to Sirt1 activation. However, in the authors’ own study, no direct changes in Sirt1 expression were observed at the proteomic level. A key suggestion for improvement is to directly attempt to measure Sirt1 activity, not just its expression, which might explain the lack of proteomic changes. If that is not feasible, the authors must more strongly emphasize and elaborate on the alternative mechanisms observed in the study, such as the influence on estrogen-dependent gene expression pathways or NF-kB/cAMP modulation, to fully substantiate the molecular basis of the radioprotection they observe. Furthermore, a deeper characterization of Protein Subset 3 is highly recommended. This third subset was uniquely affected by radiation only after long-term resveratrol pre-conditioning. Since these proteins were mainly associated with metabolism regulation and RNA and protein metabolism, not classical stress response, it complicates their direct link to radioprotection. A more detailed functional analysis and interpretation are required to explain how the disruption of this new metabolic pathway impacts the overall radioprotective phenotype. The authors should specifically investigate whether these metabolic changes are a consequence or a cause of the increased resistance. Lastly, the limitations of the cellular model must be addressed. While the focus on cardiomyocytes is justified by the problem of cardiotoxicity, a standard disclaimer about the translatability of in vitro findings to the whole organism in vivo should be added to the discussion. Further studies in living models are essential to confirm the efficacy of long-term pre-conditioning, considering the complex pharmacokinetics and metabolism of resveratrol. The authors might also clarify how the physiologically achievable concentration of 5 µM translates into a stable concentration within the heart tissue, which is a common challenge in polyphenol research.
Author Response

(The authors gave the same response as above.)

Round 2
Reviewer 2 Report
Comments and Suggestions for Authors
Dear Authors,
Thank you for your thorough and constructive revisions. In my opinion, the manuscript has been substantially improved and is suitable for publication in its current form.
Reviewer 3 Report
Comments and Suggestions for Authors
Accepted in the present form.